# Diurnal cortisol and mental well-being in middle and older age: evidence from four cohort studies

Mai Stafford,[1] Yoav Ben-Shlomo,[2] Cyrus Cooper,[3] Catharine Gale,[3,4] Michael P Gardner,[5] Marie-Claude Geoffroy,[6,7] Chris Power,[8] Diana Kuh,[1] Rachel Cooper[1]

## ABSTRACT

**Objectives** We conducted an individual participant meta-analysis to test the hypothesis that cortisol patterns indicative of dysregulated hypothalamic–pituitary–adrenal axis functioning would be prospectively associated with poorer well-being at follow-up.

**Setting** Four large UK-based cohort studies.

**Participants** Those providing valid salivary or serum cortisol samples (n=7515 for morning cortisol; n=1612 for cortisol awakening response) at baseline (age 44–82) and well-being data on the Warwick Edinburgh Mental Wellbeing Scale at follow-up (0–8 years) were included.

**Results** Well-being was not associated with morning cortisol, diurnal slope or awakening response though a borderline association with evening cortisol was found. Adjusting for sex and follow-up time, each 1 SD increase in evening cortisol was associated with a –0.47 (95% CI –1.00 to 0.05) point lower well-being. This was attenuated by adjustment for body mass index, smoking and socioeconomic position. Between-study heterogeneity was low.

**Conclusions** This study does not support the hypothesis that diurnal cortisol is prospectively associated with well-being up to 8 years later. However, replication in prospective studies with cortisol samples over multiple days is required.

## INTRODUCTION

Cortisol is a marker of hypothalamic–pituitary–adrenal (HPA) axis functioning and follows a diurnal rhythm. Large-scale epidemiological studies have measured salivary cortisol sampled several times during the course of a day to capture its rising levels during the awakening response and the subsequent decline across the day. The mental health and well-being consequences of raised cortisol levels are of interest particularly among older people given that some studies find they have higher evening cortisol levels[1] and greater total cortisol output throughout the day[2] compared with younger people though there is interindividual variation in age-related change in cortisol[3] and this may

### Strengths and limitations of this study

► Individual participant meta-analysis based on 1612–7515 participants with cortisol data.
► Prospective study with up to 8 years of follow-up.
► Validated mental well-being instrument based on 14 items capturing hedonic and eudaemonic components harmonised across cohorts.
► Salivary cortisol samples taken over 1 or 2 days, up to a maximum of 4 times per day.
► Actual waking time was not recorded in all studies.

reflect age-related change in disease rather than normal ageing.[4]

Positive mental well-being is a multidimensional concept that captures hedonic (eg, happiness and experiencing pleasure) and eudaemonic (eg, self-realisation and psychologically functioning at potential) aspects of mental health that are distinct from depressive illness.[5] Cortisol may be linked to a positive psychological state through its effect on mood-altering neurotransmitters including serotonin.[6] Cortisol also has energy-mobilising properties that may in turn promote mental well-being.[7]

Two studies of healthy older people found no association between cortisol levels and positive well-being (captured by the positive items of the General Health Questionnaire[8] or by daily positive emotions[9]). A study of women aged 65 and over found that greater eudaimonic well-being was associated with flatter cortisol slope over the day, though this was due to lower morning cortisol levels which remained low across the day.[10] All studies were small (less than 200 participants). In younger adults (ie, 55 years and younger), some studies have found that greater positive affect or happiness was associated with lower total cortisol output[11–14] but others have not.[15 16] Greater optimism has also been associated with a smaller awakening response.[12]

These studies have not examined prospective associations between baseline cortisol patterns and well-being at follow-up in large, population-based samples. Although stressors are typically associated with HPA axis activation and decreases in vagal tone, some studies suggest that there may in some cases be a subsequent response involving the dorso-vagal parasympathetic system and downregulation of the HPA axis resulting in low cortisol levels.[17] If this is the case, then a long-term inverse association between cortisol and well-being may not be evident.

The aim of the current study was to draw on individual participant data from four large British cohort studies using meta-analysis to examine the longitudinal associations between diurnal cortisol and positive mental well-being captured by an instrument which summarises positive thoughts and feelings from both hedonic and eudaemonic perspectives. We hypothesised that cortisol patterns indicative of disrupted HPA axis functioning (ie, lower early morning levels, higher evening levels, less steep decline (ie, flatter response) across the day and larger awakening response) would be associated with lower well-being at follow-up.

## METHODS
### Cohort studies
We used the four cohort studies from the Healthy Ageing across the Life Course cross-cohort research programme[18] with data on both cortisol and Warwick Edinburgh Mental Wellbeing Scale (WEMWBS): the Caerphilly Prospective Study (CaPS)[19]; the Hertfordshire Cohort Study (HCS)[20]; the Medical Research Council (MRC) National Survey of Health and Development (NSHD)[21] and the National Child Development Study (NCDS).[22] All studies received appropriate ethical approval.[19–22]

### CaPS
CaPS is a cohort of men who were recruited when they were aged 45–59 years, between 1979 and 1983, from the town of Caerphilly and adjacent villages in South Wales. In the second wave (1984–1988) the original cohort was supplemented with men of a similar age who had moved into the defined area. Salivary cortisol was assessed at phase 5 of data collection in 2000–2004 when participants were aged 65 to 82 years and well-being was assessed in 2011 when they were aged 73 to 90 years.

### HCS
HCS is a cohort of men and women born in East, North or West Hertfordshire between 1931 and 1939 whose birth and infant records were available and who were alive and still living in Hertfordshire in the 1990s. Cortisol was assessed at wave 1 of the data collection (1999–2004) when participants were aged 60–73 years and well-being was assessed in 2008 when they were aged 69–78.

### The MRC NSHD
National Survey of Health and Development (NSHD) is a nationally representative sample of people born in England, Scotland, and Wales during 1 week in March 1946 followed prospectively since birth. Cortisol and well-being data were collected in 2008–10 when cohort members were aged 62–64 years.

### The NCDS
NCDS is a nationally representative sample of people born in England, Scotland and Wales during 1 week in March 1958 followed prospectively since birth. Cortisol was assessed as part of a biomedical survey (2002–2004) when cohort members were aged 44–45 and well-being was assessed at a follow-up (2008–2009) at mean age of 50.7.

### Cortisol
All study members who had not withdrawn or been lost to follow-up at the relevant sweep were invited to participate in the cortisol sampling. In HCS, one fasting morning serum cortisol sample was taken from each participant at a research clinic between 8:30 and 9:30 (the exact timing was not recorded), frozen and subsequently measured by radioimmunoassay. Salivary cortisol samples were collected in CaPS, NSHD and NCDS at multiple times across the day with participants shown how to collect saliva using plain cotton wool swabs (salivettes) at home. Subjects were asked to chew on the salivettes for 1–2 min and a saliva sample was obtained. In CaPS, participants were requested to take samples on waking, 30 min later, at 14:00 and 22:00 on two consecutive days. In NSHD, samples were taken on waking, 30 min later and at 21:00. Samples in NCDS were taken in the first 45 min after waking and 3 hours later. Samples from CaPS, NSHD and NCDS were frozen and subsequently assayed by radio-immunoassay done at the University of Dresden which specialises in high throughput cortisol assays.[23]

### Mental well-being
Positive mental well-being was assessed using WEMWBS in all four studies. This self-completion scale captures positive affect, satisfying interpersonal relationships and positive functioning. Items are worded positively and respondents are asked to indicate how frequently, on a five-point scale, they have experienced each statement over the last 2 weeks. Statements include 'I've been feeling good about myself', 'I've been feeling close to other people', 'I've been interested in new things' and 'I've been feeling optimistic about the future'. Scores theoretically range from 14 to 70 with 70 indicating highest well-being. Where three or fewer items were missing, values were imputed based on the average score for completed items (CaPS n=43; HCS n=52; NSHD n=84; NCDS n=103). Internal consistency of the scale in all four cohorts was high (Cronbach alpha=0.91 in HCS, NSHD and NCDS and 0.93 in CaPS).[24] Validation work indicates good construct validity for a single factor structure as well

as good criterion validity and test–retest reliability and supports its use in general population samples.[25]

## Covariates

Key covariates that might confound the association between cortisol and well-being and which had been assessed in each of the cohorts were chosen a priori: sex, age at cortisol measurement, follow-up time to measurement of well-being, body mass index (BMI), smoking, and adult socio-economic position.[26–28] The covariates were measured at the same wave as cortisol samples unless otherwise stated. BMI was calculated as measured weight divided by the square of height and was categorised into quartiles when there was evidence of deviation from linearity in the association with well-being (NSHD, NCDS). Smoking was reported by participants and dichotomised into current smoker versus ex-smoker and never smoker (NCDS smoking data at age 42 years). Adult socioeconomic position was derived from own occupational class (CaPS at age 47–67, HCS at age 60–73, NSHD at age 53 or earlier if missing, NCDS at age 42) and grouped as manual or non-manual occupation (the latter indicating greater socioeconomic advantage).

## Initial treatment of the data and standardisation

Cortisol has a marked circadian rhythm and therefore the time of day at which it is sampled will affect its level. In CaPS, NSHD and NCDS actual times when the salivary samples were taken were recorded by participants. Observed values were adjusted for the time of sampling by fitting a linear or polynomial function to the association between cortisol and time of measurement and adding the resulting residuals from the best fit model to the overall mean cortisol value. This gives the estimated cortisol level at the time specified in the protocol for each participant. Morning cortisol (salivary or serum) levels were available in all four cohorts (CaPS, HCS, NSHD and NCDS) and night-time values in CaPS and NSHD. For NCDS, the later cortisol measure was taken 3 hours after the morning measure on the same day (around 11:15) and there was no evening measure. However, past publications support the notion that the diurnal decline over this shorter period is a good surrogate for the decline from morning until night,[29] and hence this measure was used to derive a measure of diurnal slope. HCS collected serum cortisol, the levels of which are around 20 times higher than the free cortisol concentrations found in saliva. However, a study on the relationship between serum and salivary cortisol in healthy individuals[30] showed that correlations were high whether taken at the same time (>0.90) or 70 min apart (0.54—0.94). Those participants who reported taking corticosteroid medication were excluded from the analysis sample (CaPS n=62; HCS n=13; NSHD n=19). In NCDS, participants were excluded from analysis if they reported taking endocrine system medication (n=396). Outlying salivary cortisol values greater than 100 nmol/L were removed (CaPS n=5; NSHD n=7; NCDS n=21), since high cortisol

values can have substantial statistical influence on estimates and it is unclear what these high values represent.[31] Morning salivary cortisol values that were not between 5:00 and noon were removed and evening values if they were before 20:00, since these participants with atypical sleeping hours may have substantially different cortisol profiles. In CaPS, cortisol values were averaged over the same measures obtained on two consecutive days. Early morning salivary cortisol was computed in CaPS and NSHD as the mean of waking and 30 min samples; to be comparable, in NCDS the cortisol measure taken within 45 min after waking was used. In HCS, morning serum cortisol was used.

To be able to combine the cohorts in a meta-analysis, the cortisol values were standardised by deriving study-specific z-scores. In NCDS, both early and late morning cortisol were positively skewed, as was night-time cortisol in CaPS and NSHD, so values were $\log_e$ transformed before they were converted to z-scores. In addition to the early morning and night-time cortisol measures, we derived the diurnal slope (CaPS, NSHD, NCDS) as early morning value subtracted from the evening (or late morning in NCDS) value and divided by the lapsed time period. The overall slope is negative and so positive z-scores indicate a flatter response. The cortisol awakening response (CAR) was derived as the difference between 30 min post-waking sample and the waking sample (CaPS, NSHD).

## Statistical methods

A two-stage meta-analysis was performed. In the first stage, we modelled WEMWBS as a function of each continuous diurnal cortisol indicator in turn using linear regression in each cohort separately, with adjustment for (1) sex, age at cortisol measurement and follow-up time to the well-being measurement, (2) additional adjustment for BMI, smoking status and socioeconomic position. There was no evidence of deviation from linearity in the association between any of the cortisol measures and WEMWBS. There was no evidence of interaction between sex and any of the cortisol measures. In sensitivity analysis, to explore possible bias arising from missing covariates, we adjusted for sex and age at cortisol measurement and follow-up time (1) using the maximum available sample with cortisol and well-being data (results not presented), (2) using the sample restricted to only those participants that had complete data on all covariates. Results did not materially differ for these two samples. In the second stage, cohort-specific estimates were pooled in random-effects meta-analyses[32] chosen a priori due to the expected heterogeneity between the different studies.

### Sensitivity analyses

We corrected regression estimates for regression dilution bias arising from error in the measurement of cortisol. The reliability ratios were estimated by regressing the cortisol measure on day 2 on the measure on day 1 from CaPS data.[33] This yielded reliability ratios of 0.554, 0.349 and 0.430 for morning, evening and slope cortisol,

**Table 1**  Descriptive statistics of participants by study

| | CaPS | HCS | NSHD | NCDS |
|---|---|---|---|---|
| Sample with WEMWBS and cortisol; n | 592 | 1055 | 1736 | 5337 |
| Gender; n (%) male | 530 (100) | 463 (44) | 809 (47) | 2464 (50) |
| Age range at cortisol measurement (years); (mean (SD)) | 65–82 (72.8 (4.0)) | 60–73 (66.6 (2.7)) | 62–64 (63.6 (0.8)) | 44–46 (44.1 (0.2)) |
| Age range at WEMWBS assessment (years); (mean (SD)) | 73–90 (80.1 (3.9)) | 69–78 (73.2 (2.4)) | 62–64 (63.6 (0.8)) | 50–51 (50.7 (0.1)) |
| Follow-up time cortisol to WEMWBS assessment (years); mean (SD) | 7.6 (0.4) | 6.6 (1.2) | 0 | 6.6 (0.3) |
| BMI (kg/m$^2$); mean(SD) | 28.2 (3.8) | 26.9 (4.2) | 27.9 (4.8) | 27.1 (4.7) |
| Adult current smoking status; n (%) smoker | 53 (11.0) | 84 (8.1) | 160 (10.2) | 893 (18.5) |
| Adult social class; n (%) manual | 303 (59.5) | 400 (39) | 499 (29.2) | 1626 (34.1) |
| WEMWBS score; mean (SD) | 53.3 (10.6) | 51.7 (8.1) | 51.8 (8.0) | 49.5 (7.9) |
| Sampling times (24 hour clock); mean | | | | |
| Waking sample | 7:37 | N/A | 7:11 | N/A |
| Waking+30 mins (+45 min for NCDS) | 8:13 | | 7:42 | 8:11 |
| Waking+3 hour 45 min | N/A | | N/A | 11:11 |
| 14:00 sample | 14:11 | | N/A | N/A |
| Evening sample | 22:00 | | 21:9.27 | N/A |
| Serum cortisol (nmol/L) | N/A | 258.1 (81.2) | N/A | N/A |
| Salivary cortisol (nmol/L) | | | | |
| Early morning; mean (SD) | 19.7 (9.5) | | 22.9 (9.6) | 21.3 (10.8) |
| Night time; median (IQR) | 2.3 (1.5, 3.5) | | 2.4 (1.7,3.7) | N/A |
| Diurnal slope (nmol/L/h); mean (SD) | −1.13 (0.7) | | −1.43 (0.7) | −4.3 (3.8) |
| Cortisol awakening response; mean (SD) | 2.3 (9.4) | | 6.4 (11.7) | N/A |
| Exclusions due to being on corticosteroids; n(%) | 62 (10%) | 13 (1.2%) | 19 (1.1%) | 396 (7.5)* |

*Endocrine medication.
CaPS, Caerphilly Prospective Study; HCS, Hertfordshire Cohort Study; NCDS, National Child Development Study; NSHD, National Survey of Health and Development; WEMWBS, Warwick Edinburgh Mental Wellbeing Scale.

respectively. In addition, because both heightened and blunted CAR have been linked to raised disease risk, we tested whether well-being differed for those with a CAR in the top third or bottom third relative to the middle third.

## RESULTS

The characteristics of the participants of the four cohorts in the analysis are shown in table 1. Age at time of well-being measurement ranged from mean (SD) age of 50.7 (0.1) years in NCDS to 80.1 (3.9) years in CaPS. Mean well-being ranged from 49.5 (7.9) in NCDS to 53.3 (10.6) in CaPS and increased with mean age of the cohort. Mean early morning cortisol values were similar for the three cohorts (CaPS, NSHD, NCDS) that had measured salivary cortisol. Night-time cortisol values were also similar although diurnal slope was more negative (indicating greater decline) in NCDS than in NSHD and CaPS.

We found no evidence of an association between early morning cortisol and well-being in the individual cohorts in sex, age and follow-up time adjusted models. The

overall pooled estimate was 0.02 (95% CI −0.17 to 0.21; p=0.8), and there was no evidence of heterogeneity in this association across studies (I$^2$=2.3; p for heterogeneity=0.4) (table 2 and figure 1). Further adjustment for all covariates did not affect the overall pooled estimate 0.01 (95% CI −0.22 to 0.24; I$^2$=18.0; p for heterogeneity=0.3).

Sex, age and follow-up time adjusted associations between evening cortisol and well-being in the individual cohorts were in the expected direction (ie, higher evening cortisol was associated with lower well-being) in NSHD: −0.33 (95% CI −0.77 to 0.11); CaPS −0.98 (95% CI −2.03 to 0.07). This indicates a weak inverse association between evening cortisol and well-being (table 2 and figure 2). Adjustment for BMI, smoking and social class attenuated this association. Again, there was no evidence of heterogeneity across studies (I$^2$=19.1; p for heterogeneity =0.3).

In the pooled analysis a flatter diurnal slope was associated with poorer well-being though this was not statistically significant before or after adjustment for all covariates

**Table 2** Overall summary estimates of effect for the associations between cortisol measures and well-being from a series of meta-analyses

| | Included cohorts | Number of individuals | Mean difference in WEMWBS score (95% CI) per SD increase in cortisol | | | | | | | |
| | | | Model 1* | | | | Model 2† | | | |
| | | | Regression coefficient (95% CI) | p Value | Tests of heterogeneity I²(%) | p Value‡ | Regression coefficient (95% CI) | p Value | Tests of heterogeneity I²(%) | p Value‡ |
|---|---|---|---|---|---|---|---|---|---|---|
| Early morning cortisol | All | 7515 | 0.02 (−0.17 to 0.21) | 0.8 | 2.3 | 0.4 | 0.01 (−0.22 to 0.24) | 0.9 | 18.0 | 0.3 |
| Evening cortisol | CaPS NSHD | 1756 | −0.47 (−1.00 to 0.05) | 0.08 | 20.3 | 0.3 | −0.31 (−0.83 to 0.21) | 0.2 | 19.1 | 0.3 |
| Flatter diurnal slope | CaPS NSHD NCDS | 6490 | −0.07 (−0.27 to 0.14) | 0.5 | 0.0 | 0.6 | −0.08 (−0.28 to 0.13) | 0.5 | 0.0 | 0.6 |
| CAR | CaPS NSHD | 1612 | −0.19 (−0.62 to 0.24) | 0.4 | 0.0 | 1.0 | −0.16 (−0.59 to 0.26) | 0.4 | 0.0 | 0.9 |

*Sex, age at cortisol assessment, follow-up time to well-being assessment.
†Model 1 plus body mass index, smoking status, adult social class.
‡p Values from Cochran's Q statistic performed as a test of between-study heterogeneity.
CaPS, Caerphilly Prospective Study; CAR, cortisol awakening response; HCS, Hertfordshire Cohort Study; NSHD, National Survey of Health and Development; NCDS, National Child Development Study; WEMWBS, Warwick Edinburgh Mental Wellbeing Scale.

(table 2 and figure 3). Results excluding NCDS (based on decline in cortisol between early and late morning) were similar (0.02 (95% CI −0.41, 0.46)). In the sex, age and follow-up time adjusted and in the fully adjusted models, a higher CAR tended to be associated with lower well-being although the association did not approach statistical significance (table 2 and figure 4).

In sensitivity analysis, correcting for regression dilution bias, the sex, age and follow-up time adjusted associations between cortisol and well-being were 0.036, −1.347 and −0.163 for morning, evening and slope cortisol, respectively. We found no evidence that well-being differed according to thirds of the CAR distribution.

## DISCUSSION

Based on meta-analysis of individual participant data from four large cohort studies, we found that morning cortisol, diurnal slope and CAR were not associated with well-being, but there was evidence that higher evening cortisol was prospectively associated with lower positive mental well-being in middle and older age. The magnitude of this association was small (−0.06 SD in well-being) in crude analysis though up to −0.17 SD in well-being accounting for possible regression dilution bias. This association was attenuated by the inclusion of BMI, smoking and socioeconomic position. It remains unclear whether obesity is secondary to HPA axis dysregulation so this may represent overadjustment for a variable on the explanatory pathway. Perceived stress is also linked to lower socioeconomic position, smoking and cortisol levels[27 30] and so isolating an association between cortisol and well-being independently of these factors needs to be interpreted with caution.

Evening cortisol is arguably the least affected by salivary sampling protocol deviations, which can bias associations between cortisol and well-being towards the null.[7] Furthermore, single-day sampling tends to bias cortisol estimates towards state rather than trait values,[31] which may also explain the lack of association with well-being up to 8 years later. Only one of the included studies captured cortisol profiles on more than 1 day and this study (ie, CaPS) only sampled cortisol on 2 days. We note that there was also a trend towards lower well-being among those with higher morning cortisol in CaPS though this did not attain statistical significance. Replication in additional studies with samples over multiple days is warranted. We examined interindividual differences in cortisol in relation to well-being up to 8 years after assessment of cortisol patterns. It is possible that an association between cortisol and well-being would only be evident over a shorter lag time. When measured on the same day, studies have found higher positive affect among those with lower cortisol output.[14 34 35] In addition, lower output in the first 45 min after waking[8] and total output across the day[10] has been associated with higher well-being over a period of 3–4 weeks. Where intensive study designs have

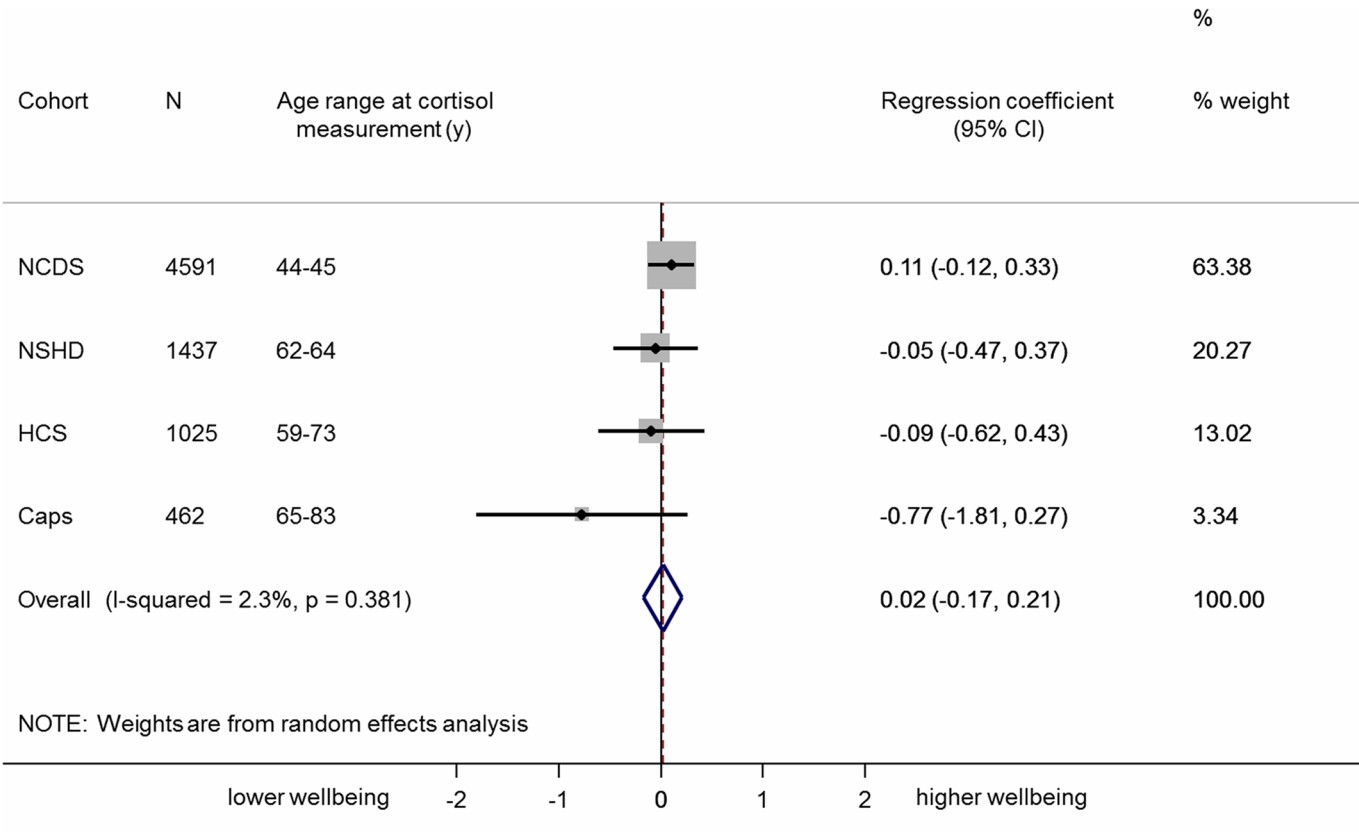

**Figure 1** Meta-analysis of the association between early morning cortisol and well-being (sex and follow-up time adjusted).

been used to measure intraindividual change based on serial measurements of both cortisol and well-being repeated over multiple days, some studies find evidence of an inverse association between cortisol output and positive affect[11 13 36–38] though others do not.[9 15 16] Trait positive affect has been found to predict higher evening

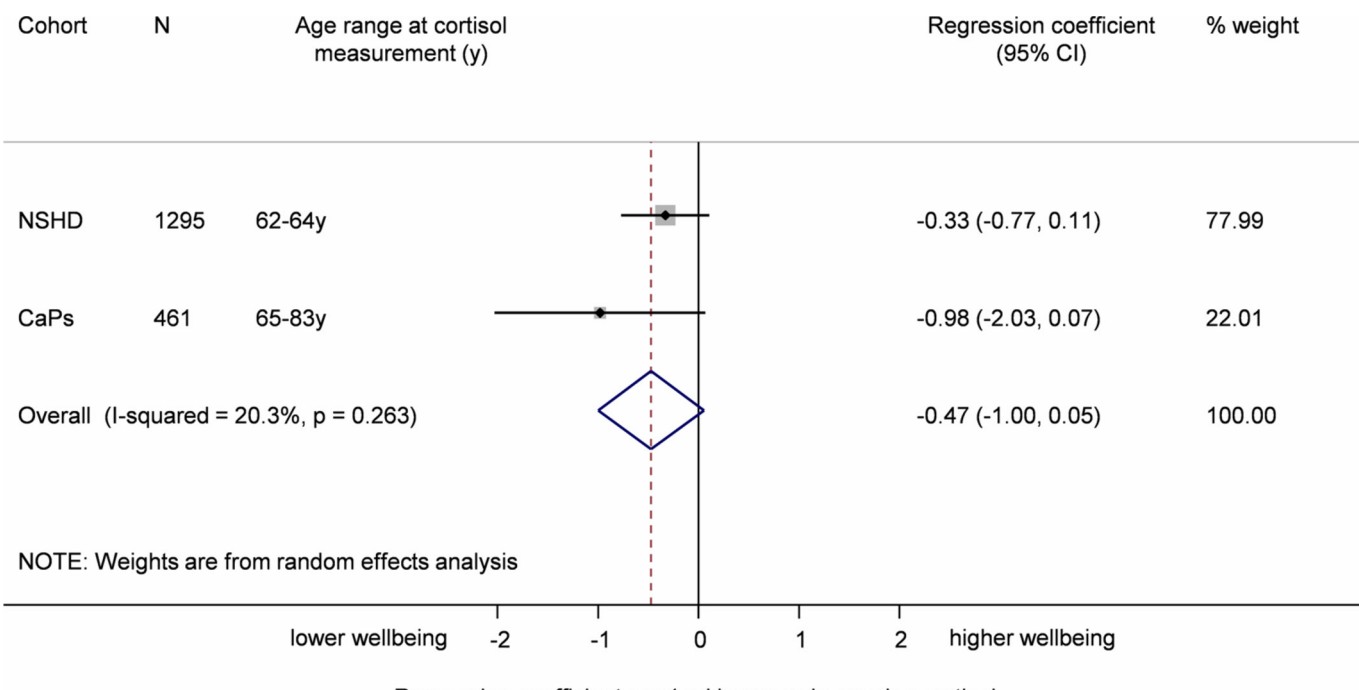

**Figure 2** Meta-analysis of the association between evening cortisol and well-being (sex and follow-up time adjusted). CaPS, Caerphilly Prospective Study; NSHD, National Survey of Health and Development.

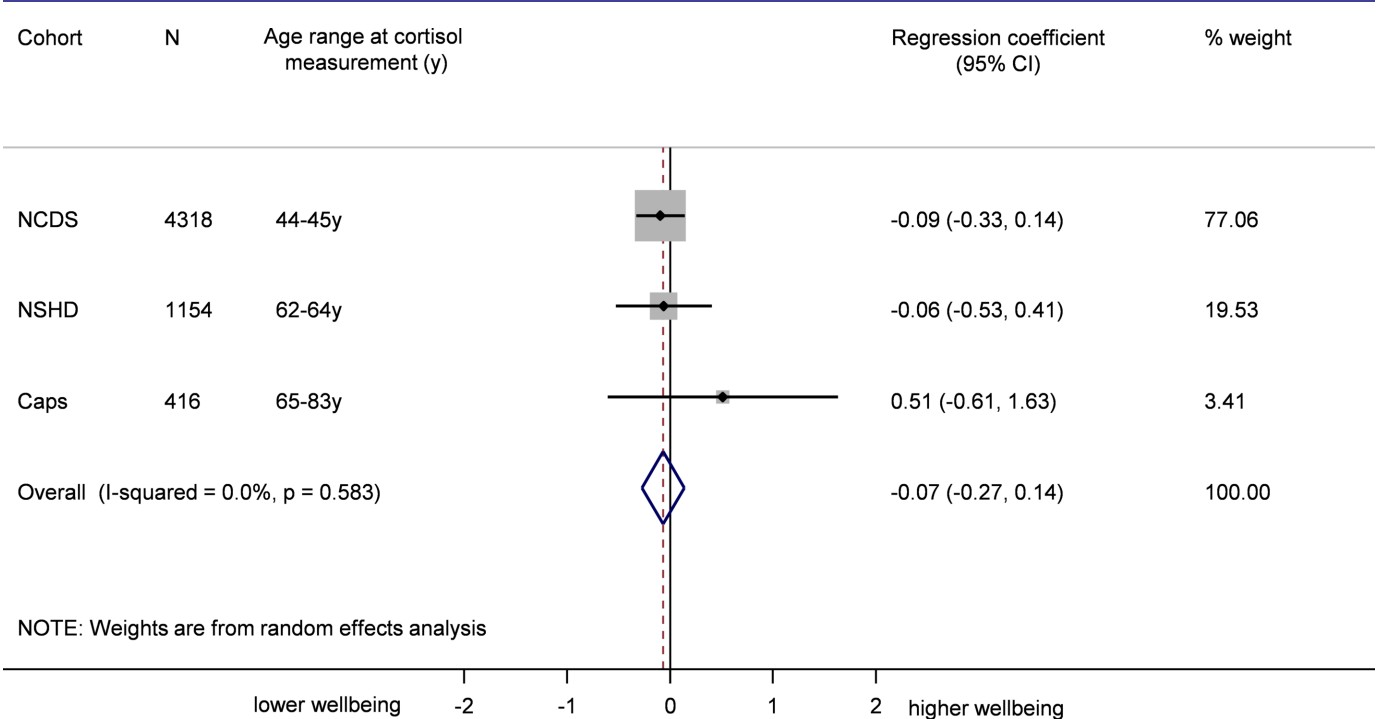

Regression coefficient per 1 sd increase in diurnal slope (flatter slope)

**Figure 3** Meta-analysis of the association between flatter diurnal slope and well-being (sex and follow-up time adjusted). CaPS, Caerphilly Prospective Study; National Child Development Study; NSHD, National Survey of Health and Development.

cortisol (in men only)[39] though we are not aware that a prospective association between baseline cortisol and subsequent mental well-being has been assessed. Future studies are warranted to examine the longitudinal association between cortisol sampled over multiple days and mental well-being to more accurately capture trait cortisol. In addition, cortisol samples taken at regular intervals throughout follow-up would enable us to

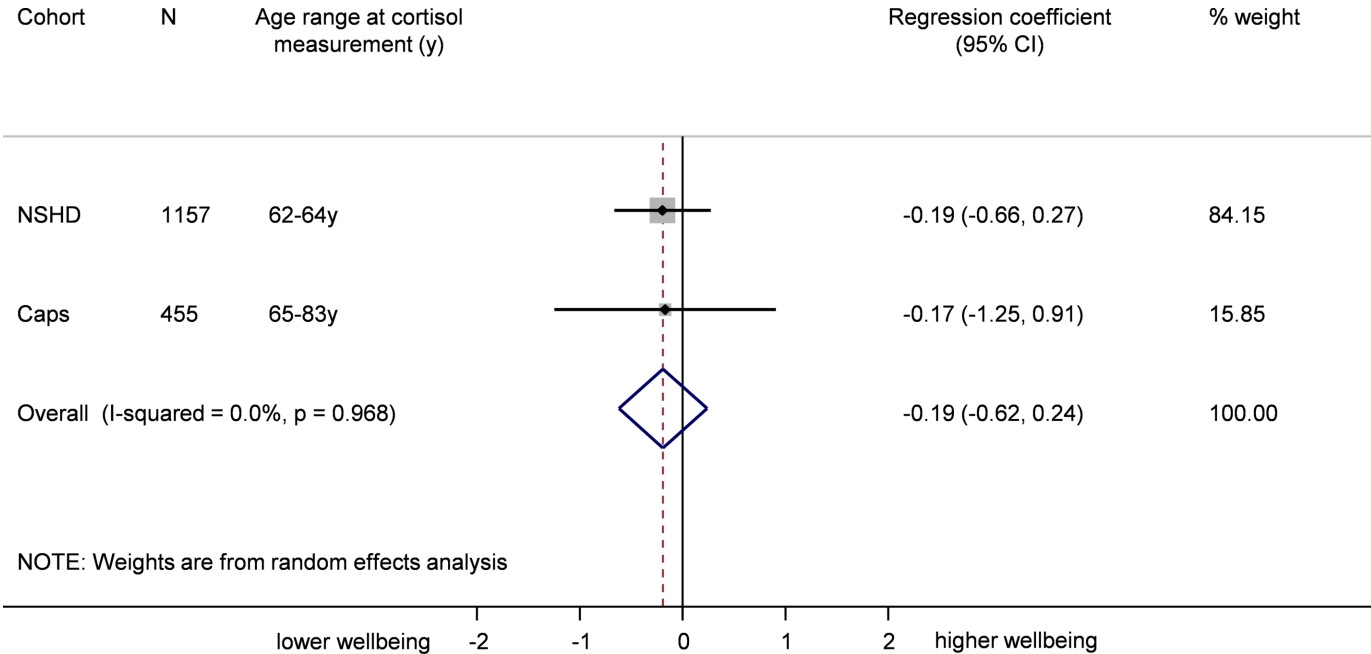

Regression coefficient per 1 sd increase in CAR

**Figure 4** Meta-analysis of the association between cortisol awakening response and well-being (sex and follow-up time adjusted). CaPS, Caerphilly Prospective Study; NSHD, National Survey of Health and Development.

identify how changes in HPA axis activation, such as hypocortisolism as a response to chronic stress, might be related to mental well-being.

Though the evidence remains inconsistent at present, explanatory pathways include the overactivation of the HPA axis which interacts with the serotonin system and may ultimately result in serotonin depletion, increasing proneness to negative emotional states and reducing positive emotionality.[6 40 41] In addition, it has been proposed that the energy-mobilising properties of cortisol may underlie an association with positive mental well-being and that this pathway may be less relevant for understanding an association between cortisol and mental ill health.[7] The evidence base is currently too limited to determine whether diurnal cortisol is related to different components of positive mental well-being and to negative mental health in the same way though there is some indication that well-being and ill-being do not show the same correlations with cortisol slope.[10] The current study used WEMWBS, designed to capture both hedonic and eudaemonic well-being over the last 2 weeks. Future research might explore a wider set of hedonic and eudaemonic components of mental well-being, as well as measures of mental ill health, in a single analytical sample to establish whether they show the same or different relationships with diurnal cortisol.

Other limitations should be acknowledged. There were differences in the protocol for the collection of cortisol. Morning serum cortisol was collected in one study and salivary cortisol in the remaining three. Among those studies with salivary cortisol, morning average cortisol was derived from samples taken at or before the typical peak (CaPS and NSHD) but after the typical peak in NCDS, and cortisol levels differ considerably during this period.[42] In addition, we excluded participants taking endocrine system medication in NCDS in contrast to participants on corticosteroid medication in the other studies though the former approach is more conservative. Despite these differences, we did not find evidence of between-study heterogeneity in the estimates from the meta-analysis (with all $I^2$ values below 21%). We used clock time but cortisol patterns are more closely anchored to waking time. Participants were instructed to provide samples at specified times postwaking but actual waking time was not recorded. In addition, a maximum of four samples per day were collected and additional measures may have provided more accurate assessment of diurnal cortisol, especially diurnal slope. We did not consider factors that may modify the prospective association between cortisol and well-being, such as social support which has been shown to buffer the health impact of stress.[43]

Nevertheless, the current study also has strengths. It includes a large number of participants (ranging from 1756 to 7515 in each analysis and considerably larger than any previous study) in population-based samples, using harmonised measures of well-being and covariates. Individual participant data meta-analysis was used, which has advantages over aggregate meta-analysis including greater statistical power and standardisation of the derivation of variables and analytical models.[44]

In summary, the findings from this meta-analysis do not provide support for the hypothesis that cortisol profiles indicative of disrupted HPA axis functioning have strong associations with positive mental well-being in healthy middle-aged and older people. Of the four diurnal cortisol levels considered here, only evening cortisol showed a prospective association with well-being and only in minimally adjusted analysis. However, cortisol was sampled on only 1 or 2 days and studies with samples across multiple days may find stronger associations if they better characterise cortisol patterns, though it seems likely that any associations, if found, will be of modest to moderate magnitude.

**Author affiliations**

[1]MRC Unit for Lifelong Health and Ageing, University College London, London, UK
[2]School of Social and Community Medicine, University of Bristol, Bristol, UK
[3]MRC Lifecourse Epidemiology Unit, University of Southampton, Southampton, UK
[4]Department of Psychology, University of Edinburgh, Edinburgh, UK
[5]Nuffield Department of Population Health, University of Oxford, Oxford, UK
[6]McGill Group for Suicide Studies at Douglas Mental Health University Institute, McGill University, Montreal, Quebec, Canada
[7]Department of Psychiatry, McGill University, Montreal, Quebec, Canada
[8]Population, Policy and Practice, UCL Great Ormond Street Institute of Child Health, University College London, London, UK

**Acknowledgements** The authors thank Stephanie Black for her contribution to data collation, statistical analysis and interpretation of the results presented here. The authors are grateful to the Centre for Longitudinal Studies (CLS), UCL Institute of Education for the use of these data and to the UK Data Service for making them available. However, neither CLS nor the UK Data Service bears any responsibility for the analysis or interpretation of these data.

**Contributors** YBS, CC, CG, CP and DK conceptualised the study. MS, YBS, MG, MCG and RC identified the variables, cleaned the data and undertook data analysis. All authors contributed to reviewing and interpreting the results, commenting on the manuscript and approved the final version.

**Funding** MS, RC and DK are supported by the UK Medical Research Council (Programme codes MRC_MC_UU_12019/1; MRC_MC_UU_12019/4; MRC_MC_UU_12019/5). The MRC National Survey of Health and Development is funded by the UK Medical Research Council. HALCyon was funded by the New Dynamics of Ageing (RES-353-25-0001) and MG was supported by this grant. Data collection of cortisol in NCDS at 45 years was funded by the United Kingdom Medical Research Council, grant G0000934. This research was supported by the National Institute for Health Research Biomedical Research Centre at Great Ormond Street Hospital for Children NHS Foundation Trust and University College London. The funders had no role in the study design, data collection, data analysis, data interpretation, writing of the report or the decision to submit the article for publication.

**Competing interests** None declared.

**Ethics approval** Ethics Committee of the Division of Medicine of the former South Glamorgan Area Health Authority (CaPS); Central Manchester Local Research Ethics Committee (NSHD); London MREC (NCDS and BCS).

**Provenance and peer review** Not commissioned; externally peer reviewed.

**Data sharing statement** MRC National Survey of Health and Development data used in this publication are available to bona fide researchers upon request to the NSHD Data Sharing Committee via a standard application procedure. Further details can be found at http://www.nshd.mrc.ac.uk/data. doi: 10.5522/NSHD/ Q101;doi:10.5522/NSHD/Q102. All requests for collaboration on the Caerphilly Prospective Study are reviewed by an independent steering committee (http://www. bris.ac.uk/social-community-medicine/projects/caerphilly/collaboration/). National Child Development Study data are available via registration with the UK Data Service. The Hertfordshire Cohort Study has governance and access arrangements that comply with MRC data sharing policy. The survey data are accessible to bonafide researchers by contacting Professor Cyrus Cooper, Director of the MRC

Lifecourse Epidemiology Unit at the University of Southampton, who can forward a collaborators' agreement.

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
