## [Reviewer comments · BMJ Open]

ARTICLE DETAILS

TITLE (PROVISIONAL)	Diurnal cortisol and mental wellbeing in middle and older age: evidence from four cohort studies
AUTHORS	Stafford, Mai; Ben-Shlomo, Yoav; Cooper, Cyrus; Gale, Catharine; Gardner, Mike; Geoffroy, Marie-Claude; Power, Christine; Kuh, Diana; Cooper, Rachel

VERSION 1 – REVIEW

REVIEWER	Sarah Stewart-Brown Warwick Medical School UK
REVIEW RETURNED	13-Feb-2017

GENERAL COMMENTS	The research described in this paper is both important and unique. There is a need for much greater understanding of the neuroendocrine underpinning of the state of mental wellbeing. The HPA axis and the hormone cortisol are likely to be involved. The authors have investigated the relationship between these two combining data from four separate cohort studies making their results robust. I have some minor comments on the methodology and discussion and some thoughts about the understanding on which the study is predicated below. My reason for suggesting that the paper is rejected are not because the paper is not valuable but because this research area is not yet sufficiently well developed for a single student such as this to be able to present a meaningful message to general medical readers. My r comments are as follows Introduction Pain avoidance is not part of definitions of hedonic wellbeing with which I am familiar. so if it is to be included in the definition in this paper it needs a specific rather than generic referencing. Methodology Smoking and over eating are both used by the general population to 'self medicate' stress. Socio-economic position influences the level of stress individuals experience. Cortisol is likely to be involved in these stresses. Adjusting analyses for these covariates is therefore arguably over adjusting. the authors appropriately present analyses with and without adjustment for covariates, but only comment on the possibility of over adjustment for BMI Intro and discussion The complexity of the physiological response to stress and the place that cortisol plays in this is not addressed. Cortisol is part of the HPA and sympathetic response. But there is another important and prevalent response to stress mediated by the dorso-vagal parasympathetic and manifested as 'freeze' or dissociation. The manifestations of this response have much in
---

	common with depression which is highly and negatively correlated with mental wellbeing. This pathway is less well researched but needs to be referred to and information on related patterns of cortisol secretion needs to be searched for and presented The study is predicated on the assumption that stress is longitudinally related to wellbeing in a consistent manor. But studies of the developmental of resilience show that stressors and the ensuing stress reaction encountered in a supportive environment lead to greater resilience and thus to improved wellbeing. Whereas similar stressors encountered in a hostile environment lead to vulnerability.s.
--	---

REVIEWER	Shireen Sindi Karolinska Institute, Sweden
REVIEW RETURNED	11-Mar-2017

GENERAL COMMENTS	The goal of this study was to investigate whether cortisol levels are associated with wellbeing at follow-up, using four UK-based cohort studies. The authors report that morning or diurnal cortisol slope were not associated with wellbeing, while a trend was found for the association between evening cortisol and wellbeing. This is an interesting study, and the main strengths are the large sample size and having the same wellbeing scale in all four studies. However, my main concern is related to the treatment of cortisol data and attempts to standardize them in this study. Cortisol measures are highly sensitive to deviation from sampling protocols and the methods chosen to treat the data. The additional attempts here to standardize the measures may limit their reliability. This unfortunately limits the conclusions that can be drawn. 1) In both the CAPS and NSHD studies, salivary cortisol was measured at the time of awakening and 30 minutes later. This protocol was designed to measure the 'cortisol awakening response (CAR)' (the increase from the awakening sample to 30 minutes later) (for a review see Stalder et al., 2016). The CAR is a key variable in this field, representing a dynamic measure that has been related to various physiological and psychological conditions. That the 2 studies had both morning measures (awakening and awakening + 30 minutes) is a strength, yet in the current study, the authors simply average the 2 samples, which causes the loss of valuable information with regards to the CAR. 2) The authors may have decided to average the two morning samples (as mentioned above) in order to combine with the data from NCDS study, where cortisol was measured within 45 minutes after awakening. However, as we know from the CAR literature, if a sample is collected at awakening, or the peak around 30 min (after a steep rise) or at 45 minutes, after a decline again, very different measures are collected. To then combine this (from the NCDS study) with the average of awakening and awakening +30 from the CAPS and NSHD studies is a crude method of combining measures for a hormone that is very sensitive to sampling time. I would advice running analyses with only the data from the CAPS and NSHD studies using the CAR measure (i.e. the dynamic increase from awakening to awakening + 30 minutes), even if this implies having a smaller sample size.
--

3) Similarly, the authors state that they also combined the aforementioned cortisol samples with the serum cortisol measures from the HCS study, and cited one study showing the high correlations between both salivary and serum cortisol levels. However, the morning serum cortisol (representing morning cortisol at one time point) which was measured hours after awakening, when participants went to the study site, is very different than the CAR measured in the CAPS and NSHD studies (representing a response to awakening). Combining them is likely to dilute potential existing associations.

4) For the CAPS and NSHD studies the diurnal slope was calculated by using the morning cortisol levels and the evening one. However, for the NCDS study, cortisol was measured in the first 45 min after awakening and 3 hours later. The authors justify using the measure 3 hours later by stating 'however past publications support the notion that the diurnal decline over this shorter period is a good surrogate for the decline from morning till night'. In fact they cite one study. The weakness of this method is that it assumes a linear decline to 3 hours and to the night cortisol levels, which is often not the case. I would suggest running the analyses with only the CAPS and NSHD studies, even if the sample size may be smaller.

5) It is unclear why only in the NCDS study, participants were excluded from analyses if they reported taking endocrine system medications, and not from the other studies.

6) It is also unclear how the cut-off for salivary cortisol values was decided (100 nmol/L)

7) The authors state 'Morning salivary cortisol values that were not between 5am and noon were removed and evening values if they were before 8pm, since these participants may be shiftworkers with substantially different cortisol profiles'. Is there information available on whether or not individuals were shiftworkers? What proportion were they in the different populations? And as the authors state, since they have very different cortisol profiles, it would be important to perform sensitivity analyses with and without their values.

8) In the HCS study, it is unclear what the time range and mean (SD) was for the fasting morning serum cortisol

9) It is unclear how participants were selected to participate in the cortisol sampling portions of the different studies

10) In the introduction, the authors state 'The mental health and wellbeing consequences of raised cortisol levels are of interest particularly among older people given that they have higher evening cortisol levels'. The references provided are from 1996 and 2006. And indeed, this was believed to be the case. However, more recently it has been argued and shown that higher cortisol levels may indicate underlying pathology, whereas among healthy older adults, they do not show elevations in evening cortisol as a function of aging per se (see Souza-Talarico et al., 2011; Sindi et al., 2013).

11) Regarding socioeconomic position, it was derived from own occupational class and grouped as manual or non-manual occupation. The limitation with this method is for example if women had manual occupations, yet their husbands had non-manual occupations, allowing for a high household income,

then this would be an inaccurate measure. Can this variable be improved?

12) Why was education not adjusted for?

13) Under 'initial treatment of the data and standardization', the authors state "Observed values were adjusted for the time of sampling by fitting a linear or polynomial function to the association between cortisol and time of measurement and adding the resulting residuals from the best fit model to the overall mean cortisol value. This gives the estimated cortisol level at the time specified in the protocol for each participant". Does this method take into account the consequences of deviation from sampling times, which are specific to the scheduled sampling time? What I mean is for example, a 15 min delay from awakening will lead to an over-estimation of cortisol levels (because cortisol increases after awakening). In contrast, a 15 min delay for the awakening +30 minutes sample would lead to an under-estimation of cortisol levels (because cortisol decreases following the peak at awakening +30 minutes)

14) The descriptions of the different studies need elaboration. For example the follow-up time (range, mean and SD), participation rates, the mean age and SD at the different measurement time points, the numbers of participants in the main cohort study and those with measures available for the current study. Also, references are needed for previous publications where these cohort studies were described in more detail.

15) Under 'statistical methods', it states "cohort-specific estimates were pooled in random-effects meta-analyses". However, the reference (#32) is an article titled 'Meta-analysis in clinical trials.' How is this reference on clinical trials relevant for the current study with cohort studies?

16) Under results, the authors comment on the similarity between morning cortisol levels, but nothing is mentioned about the serum cortisol, night cortisol, or the slope.

17) When the authors state 'there was no evidence of heterogeneity across studies', it is unclear, heterogeneity in what?

18) In Table 1, information is missing on the mean and SD under 'age range at cortisol measurement', similarly to how it was done for the WEMWBS row

19) In Table 1, information is needed on follow-up time (range, mean, SD)

20) There is no information on participant consent, ethics approval etc

21) The following statement is currently under 'Initial treatment of the data and standardization', "In CaPS, NSHD and NCDS actual times when the salivary samples were taken were recorded by participants", instead it should be under 'Cortisol'.

22) Minor comment: If the authors use the acronym BMI for body mass index, they should use it consistently in the manuscript.

	References: Sindi, S., Fiocco, A. J., Juster, R. P., Pruessner, J., & Lupien, S. J. (2013). When we test, do we stress? Impact of the testing environment on cortisol secretion and memory performance in older adults. Psychoneuroendocrinology, 38(8), 1388-1396. Stalder, T., Kirschbaum, C., Kudielka, B. M., Adam, E. K., Pruessner, J. C., Wüst, S., ... & Miller, R. (2016). Assessment of the cortisol awakening response: expert consensus guidelines. Psychoneuroendocrinology, 63, 414-432. Souza-Talarico, J. N. D., Marin, M. F., Sindi, S., & Lupien, S. J. (2011). Effects of stress hormones on the brain and cognition: evidence from normal to pathological aging. Dementia & Neuropsychologia, 5(1), 8-16.
--	--

VERSION 1 – AUTHOR RESPONSE

Comments from reviewer #1:

The research described in this paper is both important and unique. There is a need for much greater understanding of the neuroendocrine underpinning of the state of mental wellbeing. The HPA axis and the hormone cortisol are likely to be involved. The authors have investigated the relationship between these two combining data from four separate cohort studies making their results robust.

Our response: We thank the reviewer for their positive overall assessment of our paper.

I have some minor comments on the methodology and discussion and some thoughts about the understanding on which the study is predicated below. My reason for suggesting that the paper is rejected are not because the paper is not valuable but because this research area is not yet sufficiently well developed for a single student such as this to be able to present a meaningful message to general medical readers.

Our response: We are not sure how to interpret this comment but would like to highlight that the paper is based on data from 4 cohort studies with >1700 observations for each analysis (and up to 7500 observations). Our conclusion explicitly states that we tested an association between diurnal cortisol and wellbeing over 8 years of follow-up (thus we do not rule out significant associations over shorter periods). We also note there that replication studies are warranted particularly because of the lack of multiple cortisol samples.

Introduction: Pain avoidance is not part of definitions of hedonic wellbeing with which I am familiar. so if it is to be included in the definition in this paper it needs a specific rather than generic referencing.

Our response: We agree and have removed “pain avoidance” from the text. We now refer to “happiness and experiencing pleasure” (page 4).

Methodology: Smoking and over eating are both used by the general population to 'self medicate' stress. Socio-economic position influences the level of stress individuals experience. Cortisol is likely to be involved in these stresses. Adjusting analyses for these covariates is therefore arguably over adjusting. The authors appropriately present analyses with and without adjustment for covariates, but only comment on the possibility of over adjustment for BMI.

Our response: Thank you for raising this important point. We agree that perceived stress is likely to be implicated and may affect cortisol levels and behaviours such as smoking and over-eating.

Although consideration of the upstream determinants of cortisol was not a focus of this paper, we have added a sentence to page 13 to briefly acknowledge the challenge: “Perceived stress is also linked to lower socioeconomic position, smoking and cortisol levels [27, 30] and so isolating an association between cortisol and wellbeing independently of these factors needs to be interpreted with caution.”

Intro and discussion: The complexity of the physiological response to stress and the place that cortisol plays in this is not addressed. Cortisol is part of the HPA and sympathetic response. But there is another important and prevalent response to stress mediated by the dorso-vagal parasympathetic and manifested as 'freeze' or dissociation. The manifestations of this response have much in common with depression which is highly and negatively correlated with mental wellbeing. This pathway is less well researched but needs to be referred to and information on related patterns of cortisol secretion needs to be searched for and presented.

Our response: Thank you for this suggestion. We have added the following sentences on page 5 to acknowledge that we may not see a prospective association over this length of follow-up because of alterations to HPA-axis activation: “Although stressors are typically associated with HPA axis activation and decreases in vagal tone, some studies suggest there may in some cases be a subsequent response involving the dorso-vagal parasympathetic system and down-regulation of the HPA axis resulting in low cortisol levels (new reference added Porges 2009). If this is the case then a long-term inverse association between cortisol and wellbeing may not be evident.”

Porges SW. The polyvagal theory: New insights into adaptive reactions of the autonomic nervous system. *Cleve Clin J Med* 2009; 76(Suppl 2):S86–S90

On page 14 we have added the following: “In addition, cortisol samples taken at regular intervals through follow-up would enable us to identify how changes in HPA-axis activation, such as hypocortisolism as a response to chronic stress, might be related to mental wellbeing.”

The study is predicated on the assumption that stress is longitudinally related to wellbeing in a consistent manor. But studies of the developmental of resilience show that stressors and the ensuing stress reaction encountered in a supportive environment lead to greater resilience and thus to improved wellbeing. Whereas similar stressors encountered in a hostile environment lead to vulnerability.

Our response: The reviewer raises the possibility that the association between cortisol and wellbeing may be moderated by environmental factors such as support or hostility. We have added a sentence to the limitations section of the discussion to acknowledge this: “We did not consider factors that may modify the prospective association between cortisol and wellbeing, such as social support which has been shown to buffer the health impact of stress (new reference added Hostinar 2015).”

Hostinar CE, Gunnar MR. Social Support Can Buffer against Stress and Shape Brain Activity. *AJOB Neurosci* 2015;6:34-42.

Comments from reviewer #2:

Comment: This is an interesting study, and the main strengths are the large sample size and having the same wellbeing scale in all four studies. However, my main concern is related to the treatment of cortisol data and attempts to standardize them in this study. Cortisol measures are highly sensitive to deviation from sampling protocols and the methods chosen to treat the data. The additional attempts here to standardize the measures may limit their reliability. This unfortunately limits the conclusions that can be drawn.

Our response: We thank the reviewer for noting the strengths of our study. We agree that there are challenges in harmonising data in order to undertake meta-analyses. We would like to note though that we have published meta-analysis of these cortisol data previously (Gardner et al 2013) and that we have been explicit about the decisions we made when cleaning the cortisol data. Our treatment of the cortisol data is typical of that for large, community-based studies (removing outliers and those on relevant medication, correcting for deviations from protocol in the timing of the sample) and we do not consider that this was influenced by standardising across the four cohorts. We justify our choice for each cortisol profile indicator included in the responses below. The figures show results from each individual cohort for completeness.

Gardner MP, Lightman S, Sayer AA, Cooper C, Cooper R, Deeg D, Ebrahim S, Gallacher J, Kivimaki M, Kumari M, Kuh D, Martin RM, Peeters G, Ben-Shlomo Y; Halcyon Study Team. Dysregulation of the hypothalamic pituitary adrenal (HPA) axis and physical performance at older ages: an individual participant meta-analysis. *Psychoneuroendocrinology*. 2013;38:40-9.

1) In both the CAPS and NSHD studies, salivary cortisol was measured at the time of awakening and 30 minutes later. This protocol was designed to measure the 'cortisol awakening response (CAR)' (the increase from the awakening sample to 30 minutes later) (for a review see Stalder et al., 2016). The CAR is a key variable in this field, representing a dynamic measure that has been related to various physiological and psychological conditions. That the 2 studies had both morning measures (awakening and awakening + 30 minutes) is a strength, yet in the current study, the authors simply average the 2 samples, which causes the loss of valuable information with regards to the CAR.

Our response: Thank you for this suggestion. We originally did look at the CAR but elected not to include it in this paper because it is particularly sensitive to sample timing which varied between studies. In addition, previous evidence indicates that both high and low CAR could be suboptimal. As an aside, we found no evidence of an association between CAR and wellbeing (please see figure below).

2) The authors may have decided to average the two morning samples (as mentioned above) in order to combine with the data from NCDS study, where cortisol was measured within 45 minutes after awakening. However, as we know from the CAR literature, if a sample is collected at awakening, or the peak around 30 min (after a steep rise) or at 45 minutes, after a decline again, very different measures are collected. To then combine this (from the NCDS study) with the average of awakening and awakening +30 from the CAPS and NSHD studies is a crude method of combining measures for a hormone that is very sensitive to sampling time. I would advise running analyses with only the data from the CAPS and NSHD studies using the CAR measure (i.e. the dynamic increase from awakening to awakening + 30 minutes), even if this implies having a smaller sample size.

Our response: Please see response to 1). We note that Table 1 shows mean morning cortisol (based on averages of awakening and awakening + 30 mins) that are similar in the 3 birth cohorts (19.7, 22.9 and 21.3 respectively). The peak is typically measured at 30-45 minutes post-waking (Adam & Kumari 2009).

Adam EK, Kumari M. Assessing salivary cortisol in large-scale, epidemiological research. *Psychoneuroendocrinology* 2009;34:1423-36.

3) Similarly, the authors state that they also combined the aforementioned cortisol samples with the serum cortisol measures from the HCS study, and cited one study showing the high correlations between both salivary and serum cortisol levels. However, the morning serum cortisol (representing morning cortisol at one time point) which was measured hours after awakening, when participants went to the study site, is very different than the CAR measured in the CAPS and NSHD studies (representing a response to awakening). Combining them is likely to dilute potential existing associations.

Our response: We agree, it is possible that including HCS based on serum cortisol (and NCDS based on awakening + 45 mins) would dilute associations. However, as can be seen in Figure 1A, there is no suggestion of an association between morning cortisol and wellbeing in NCDS, NSHD or HCS. There is a suggestion of an inverse association in CAPS. This is the one study that sampled cortisol over two days and so we make the recommendation that replication is needed with studies which have samples over multiple days.

4) For the CAPS and NSHD studies the diurnal slope was calculated by using the morning cortisol levels and the evening one. However, for the NCDS study, cortisol was measured in the first 45 min after awakening and 3 hours later. The authors justify using the measure 3 hours later by stating 'however past publications support the notion that the diurnal decline over this shorter period is a good surrogate for the decline from morning till night'. In fact they cite one study. The weakness of this method is that it assumes a linear decline to 3 hours and to the night cortisol levels, which is often not the case. I would suggest running the analyses with only the CAPS and NSHD studies, even if the sample size may be smaller.

Our response: As Figure 1C shows, the estimate of the association between diurnal slope and wellbeing was not materially different in NCDS and the other two studies included in that analysis. To address this comment, we have now repeated the meta-analysis including only NSHD and CAPS and we report this result in the text (page 12): "Results excluding NCDS (based on decline in cortisol between early and late morning) were similar (0.02 (95% CI -0.41, 0.46))."

5) It is unclear why only in the NCDS study, participants were excluded from analyses if they reported taking endocrine system medications, and not from the other studies.

Our response: We have taken the more conservative approach to exclude NCDS participants on endocrine system medication because we did not have access to more detailed data on corticosteroid medication in this analysis.

6) It is also unclear how the cut-off for salivary cortisol values was decided (100 nmol/L)

Our response: Examination of the salivary cortisol distribution indicated that values >100 nmol/L were clear outliers.

7) The authors state 'Morning salivary cortisol values that were not between 5am and noon were removed and evening values if they were before 8pm, since these participants may be shiftworkers with substantially different cortisol profiles'. Is there information available on whether or not individuals were shiftworkers? What proportion were they in the different populations? And as the authors state, since they have very different cortisol profiles, it would be important to perform sensitivity analyses with and without their values.

Our response: On reflection, given the age range of the cohort members, shiftwork is unlikely to be the primary explanation for morning values outside of 5am to noon and evening values before 8pm. We have edited the sentence on page 9 so that it now reads: "since these participants with atypical sleeping hours may be shift-workers with may have substantially different cortisol profiles." Numbers were small (e.g. n=51 observations in NSHD) and so statistical power is too low to examine this group separately.

8) In the HCS study, it is unclear what the time range and mean (SD) was for the fasting morning serum cortisol

Our response: We now clarify that the fasting morning serum cortisol sample was taken at a research clinic between 8.30 and 9.30 AM.

9) It is unclear how participants were selected to participate in the cortisol sampling portions of the different studies.

Our response: All those who had not withdrawn or been lost to follow-up at the relevant sweep were invited to participate in the cortisol sampling. This information has been added (page 7).

10) In the introduction, the authors state 'The mental health and wellbeing consequences of raised cortisol levels are of interest particularly among older people given that they have higher evening cortisol levels'. The references provided are from 1996 and 2006. And indeed, this was believed to be the case. However, more recently it has been argued and shown that higher cortisol levels may indicate underlying pathology, whereas among healthy older adults, they do not show elevations in evening cortisol as a function of aging per se (see Souza-Talarico et al., 2011; Sindi et al., 2013).

Our response: Thank you for raising this. We have now clarified that there is inter-individual variation in how cortisol changes with age and the potential role of underlying pathology. The text (page 4) now states: “The mental health and wellbeing consequences of raised cortisol levels are of interest particularly among older people given that some studies find they have higher evening cortisol levels[1] and greater total cortisol output throughout the day[2] compared with younger people though there is inter-individual variation in age-related change in cortisol [3] and this may reflect age-related change in disease rather than normal ageing (new reference added Souza-Talarico et al. 2011).” Souza-Talarico, J. N. D., Marin, M. F., Sindi, S., & Lupien, S. J. (2011). Effects of stress hormones on the brain and cognition: evidence from normal to pathological aging. *Dementia & Neuropsychologia*, 5(1), 8-16.

11) Regarding socioeconomic position, it was derived from own occupational class and grouped as manual or non-manual occupation. The limitation with this method is for example if women had manual occupations, yet their husbands had non-manual occupations, allowing for a high household income, then this would be an inaccurate measure. Can this variable be improved?

Our response: We checked whether own occupational class or head of household class was more closely related to cortisol and found a stronger association for the latter.

12) Why was education not adjusted for?

Our response: We found no evidence of an association between education and cortisol when occupational class was adjusted for and so education did not meet the formal definition of a confounder.

13) Under ‘initial treatment of the data and standardization’, the authors state “Observed values were adjusted for the time of sampling by fitting a linear or polynomial function to the association between cortisol and time of measurement and adding the resulting residuals from the best fit model to the overall mean cortisol value. This gives the estimated cortisol level at the time specified in the protocol for each participant”. Does this method take into account the consequences of deviation from sampling times, which are specific to the scheduled sampling time? What I mean is for example, a 15 min delay from awakening will lead to an over-estimation of cortisol levels (because cortisol increases after awakening). In contrast, a 15 min delay for the awakening +30 minutes sample would lead to an under-estimation of cortisol levels (because cortisol decreases following the peak at awakening +30 minutes)

Our response: Yes, this approach accounts for the deviation from specific scheduled sampling time. The polynomial function allows for an average increase in cortisol with time before the specified time and an average decrease with time after the specified time, for example.

14) The descriptions of the different studies need elaboration. For example the follow-up time (range, mean and SD), participation rates, the mean age and SD at the different measurement time points, the numbers of participants in the main cohort study and those with measures available for the current study. Also, references are needed for previous publications where these cohort studies were described in more detail.

Our response: We have added the ranges, means and SD as suggested to Table 1. These cohort studies are described in more detail in the cohort profile papers (references 19-22).

15) Under 'statistical methods', it states "cohort-specific estimates were pooled in random-effects meta-analyses". However, the reference (#32) is an article titled 'Meta-analysis in clinical trials.' How is this reference on clinical trials relevant for the current study with cohort studies?

Our response: The random-effects meta-analysis approach is a well-accepted way of combining estimates of effect size across several studies and it is used for both randomised and non-randomised designs. (Here the "random-effects" term refers to the way that the estimates are combined across studies rather than to the original study design.) The authors of the paper we cited note this on page 186 of their paper. This is the standard reference for this approach. DerSimonian R, Laird N. Meta-analysis in clinical trials. *Control Clin Trials* 1986;7:177-88.

16) Under results, the authors comment on the similarity between morning cortisol levels, but nothing is mentioned about the serum cortisol, night cortisol, or the slope.

Our response: We now note that night time cortisol was also similar but that there was some difference in slope for NCDS versus the other studies (page 11). As noted above, we now include analysis of diurnal slope with and without NCDS (page 12).

17) When the authors state 'there was no evidence of heterogeneity across studies', it is unclear, heterogeneity in what?

Our response: We now clarify that there was no evidence of heterogeneity in the association between early morning cortisol and wellbeing across the studies (page 11).

18) In Table 1, information is missing on the mean and SD under 'age range at cortisol measurement', similarly to how it was done for the WEMWBS row

Our response: We have now added mean and SD for age at cortisol measurement to Table 1.

19) In Table 1, information is need on follow-up time (range, mean, SD)

Our response: Mean and SD follow-up time has been added to Table 1.

20) There is no information on participant consent, ethics approval etc.

Our response: We now note that ethical approval was obtained for each of these studies. Rather than list the relevant ethics committee for each one, we refer again to the cohort profiles which describe these studies and the approval in more detail.

21) The following statement is currently under 'Initial treatment of the data and standardization', "In CaPS, NSHD and NCDS actual times when the salivary samples were taken were recorded by participants", instead it should be under 'Cortisol'.

Our response: Thank you for the suggestion but as this sentence is relevant for explaining how we derived cortisol level at the per protocol sampling time (rather than at the observed sampling time), we think it fits more appropriately in the section on initial treatment of the data.

22) Minor comment: If the authors use the acronym BMI for body mass index, they should use it consistently in the manuscript.

Our response: We have replaced “body mass index” with BMI except for the first occurrence and where it appears in the Tables.

VERSION 2 – REVIEW

REVIEWER	Shireen Sindi Karolinska Institute
REVIEW RETURNED	30-May-2017

GENERAL COMMENTS	wellbeing at follow-up, using four UK-based cohort studies. The authors report that morning or diurnal cortisol slope were not associated with wellbeing, while a trend was found for the association between evening cortisol and wellbeing. This is an interesting study, and the main strengths are the large sample size and having the same wellbeing scale in all four studies. However, my main concern is related to the treatment of cortisol data and attempts to standardize them in this study. Cortisol measures are highly sensitive to deviation from sampling protocols and the methods chosen to treat the data. The additional attempts here to standardize the measures may limit their reliability. This unfortunately limits the conclusions that can be drawn. 1) In both the CAPS and NSHD studies, salivary cortisol was measured at the time of awakening and 30 minutes later. This protocol was designed to measure the ‘cortisol awakening response (CAR)’ (the increase from the awakening sample to 30 minutes later) (for a review see Stalder et al., 2016). The CAR is a key variable in this field, representing a dynamic measure that has been related to various physiological and psychological conditions. That the 2 studies had both morning measures (awakening and awakening + 30 minutes) is a strength, yet in the current study, the authors simply average the 2 samples, which causes the loss of valuable information with regards to the CAR. Indeed I agree that both a high and low CAR is suboptimal. 2) The authors may have decided to average the two morning samples (as mentioned above) in order to combine with the data from NCDS study, where cortisol was measured within 45 minutes after awakening. However, as we know from the CAR literature, if a sample is collected at awakening, or the peak around 30 min (after a steep rise) or at 45 minutes, after a decline again, very different measures are collected. To then combine this (from the NCDS study) with the
--

average of awakening and awakening +30 from the CAPS and NSHD studies is a crude method of combining measures for a hormone that is very sensitive to sampling time. I would advice running analyses with only the data from the CAPS and NSHD studies using the CAR measure (i.e. the dynamic increase from awakening to awakening + 30 minutes), even if this implies having a smaller sample size. In the filed of psychoneuroendocrinology, the reason for having 2 measures (awakening, awakening+30 minutes) or with an additional third measure (awakening+45 minutes) is to measure the dynamic cortisol changes during this period.

The authors state that the sample averages are the same across studies. However, the averages omits important information. Just as a hypothetical example, participant (A) may have an awakning cortisol at 10 nmol/L, which increases to 30 nmol/L at the awakening+30 measure. His average is 20 nmol/L. Participant (B) may have an awakning cortisol at 20 nmol/L, which remains flat at 20 nmol/L at the awakening+30 measure. They both have an average of 20 nmol/L, but participant (A) has a healthy profile, while participant (B) is suffering from burnout, which is why he has a flat profile. This would clearly not be capture by using an average.

The authors state that the objective of the study is “to test the hypothesis that cortisol patterns indicative of dysregulated HPA-axis functioning would be prospectively associated with poorer wellbeing at follow-up”. However, an average of the 2 morning cortisol measures (e.g. awakening and awakening +30) will certainly not capture dysregulations. Moreover, using the average contradicts what the authors report in the introduction “Large-scale epidemiological studies have measured salivary cortisol sampled several times during the course of a day to capture its rising levels during the awakening response”. Again, by using an average, this ‘rise’ cannot be captured, and important information is lost.

A measure of change in cortisol is used in this study for the evening proportion, but it is unclear why it is not also used for the morning. Also, whether a measure is taken at awakening+30 or awakning+45 does make a difference. Usually +30 is the peak, while +45 is the decline following the peak. In order to make it easier for the authors to visualize the point I am making, I’m attaching some examples from previous articles in the field.

Vreeburg, S. A., Hoogendijk, W. J., van Pelt, J., DeRijk, R. H., Verhagen, J. C., van Dyck, R., ... & Penninx, B. W. (2009). Major depressive disorder and hypothalamic-pituitary-adrenal axis activity: results from a large cohort study. *Archives of general psychiatry*, 66(6), 617-626.

- ◆— Personal diagnosis ($n = 1262$)
- Diagnosed parental history ($n = 74$)
- ▲— Self-reported parental history ($n = 114$)
- Control without parental history ($n = 180$)

Ulrike, S., Reinhold, L., & Dirk, H. (2013). Major depression in young girls is related to altered cortisol awakening response. *European child & adolescent psychiatry*, 22(6), 379-384.

3) Similarly, the authors state that they also combined the aforementioned cortisol samples with the serum cortisol measures from the HCS study, and cited one study showing the high correlations between both salivary and serum cortisol levels. However, the morning serum cortisol (representing morning cortisol at one time point) which was measured hours after awakening, when participants went to the study site, is very different than the CAR measured in the CAPS and NSHD studies (representing a response to awakening). Combining them is likely to dilute potential existing associations. This can be mentioned in the discussion.

4) For the CAPS and NSHD studies the diurnal slope was calculated by using the morning cortisol levels and the evening one. However, for the NCDS study, cortisol was measured in the first 45 min after awakening and 3 hours later. The authors justify using the measure 3 hours later by stating 'however past publications support the notion that the diurnal decline over this shorter period is a good surrogate for the decline from morning till night'. In fact they cite one study. The weakness of this method is that it assumes a linear decline to 3 hours and to the night cortisol levels, which is often not the case. I would suggest running the analyses with only the CAPS and NSHD studies, even if the sample size may be smaller.

5) It is unclear why only in the NCDS study, participants were excluded from analyses if they reported taking endocrine system medications, and not from the other studies. Did you mean 'in this analysis' or from this dataset? This should be mentioned as a limitation.

6) It is also unclear how the cut-off for salivary cortisol values was decided (100 nmol/L)

7) The authors state 'Morning salivary cortisol values that were not between 5am and noon were removed and evening values if they were before 8pm, since these participants may be shiftworkers with substantially different cortisol profiles'. Is there information available on whether or not individuals were shiftworkers?

What proportion were they in the different populations? And as the authors state, since they have very different cortisol profiles, it would be important to perform sensitivity analyses with and without their values.

This modified sentence is unclear (grammar error). Do the results change if these 51 are excluded?

8) In the HCS study, it is unclear what the time range and mean (SD) was for the fasting morning serum cortisol

	9) It is unclear how participants were selected to participate in the cortisol sampling portions of the different studies 10) In the introduction, the authors state 'The mental health and wellbeing consequences of raised cortisol levels are of interest particularly among older people given that they have higher evening cortisol levels'. The references provided are from 1996 and 2006. And indeed, this was believed to be the case. However, more recently it has been argued and shown that higher cortisol levels may indicate underlying pathology, whereas among healthy older adults, they do not show elevations in evening cortisol as a function of aging per se (see Souza-Talarico et al., 2011; Sindi et al., 2013). Thank you for adding this important point. 11) Regarding socioeconomic position, it was derived from own occupational class and grouped as manual or non-manual occupation. The limitation with this method is for example if women had manual occupations, yet their husbands had non-manual occupations, allowing for a high household income, then this would be an inaccurate measure. Can this variable be improved?. That is interesting. Did you incorporate it in the analyses? 12) Why was education not adjusted for? 13) Under 'initial treatment of the data and standardization', the authors state "Observed values were adjusted for the time of sampling by fitting a linear or polynomial function to the association between cortisol and time of measurement and adding the resulting residuals from the best fit model to the overall mean cortisol value. This gives the estimated cortisol level at the time specified in the protocol for each participant". Does this method take into account the consequences of deviation from sampling times, which are specific to the scheduled sampling time? What I mean is for example, a 15 min delay from awakening will lead to an over-estimation of cortisol levels (because cortisol increases after awakening). In contrast, a 15 min delay for the awakening +30 minutes sample would lead to an under-estimation of cortisol levels (because cortisol decreases following the peak at awakening +30 minutes) 14) The descriptions of the different studies need elaboration. For example the follow-up time (range, mean and SD), participation rates, the mean age and SD at the different measurement time points, the numbers of participants in the main cohort study and those with measures available for the current study. Also,
--	---

	references are needed for previous publications where these cohort studies were described in more detail. 15) Under 'statistical methods', it states "cohort-specific estimates were pooled in random-effects meta-analyses". However, the reference (#32) is an article titled 'Meta-analysis in clinical trials.' How is this reference on clinical trials relevant for the current study with cohort studies? 16) Under results, the authors comment on the similarity between morning cortisol levels, but nothing is mentioned about the serum cortisol, night cortisol, or the slope. Thank you for adding this information. 17) When the authors state 'there was no evidence of heterogeneity across studies', it is unclear, heterogeneity in what? 18) In Table 1, information is missing on the mean and SD under 'age range at cortisol measurement', similarly to how it was done for the WEMWBS row 19) In Table 1, information is need on follow-up time (range, mean, SD) 20) There is no information on participant consent, ethics Thank you for adding this information. It is also standard for many journals to request for a statement that all participants provided written informed consent. 21) The following statement is currently under 'Initial treatment of the data and standardization', "In CaPS, NSHD and NCDS actual times when the salivary samples were taken were recorded by participants", instead it should be under 'Cortisol'. 22) Minor comment: If the authors use the acronym BMI for body mass index, they p.10 It states, 'we adjusted for sex and age at cortisol...', do you mean , age at cortisol 'measurements'? p.13 The last part of this sentence needs revision, it is currently unclear: 'Only one of the included studies captured cortisol profiles on more than one day and this study only on two days'. Warwick Edinburgh Mental Wellbeing Scale. The abbreviation was not defined before its first use in the abstract. In the manuscript, the abbreviation should be defined at its first appearance (p.5), and then used consistently across the manuscript (e.g. p.7 &14).
--	--

VERSION 2 – AUTHOR RESPONSE

Comment: This is an interesting study, and the main strengths are the large sample size and having the same wellbeing scale in all four studies. However, my main concern is related to the treatment of cortisol data and attempts to standardize them in this study. Cortisol measures are highly sensitive to deviation from sampling protocols and the methods chosen to treat the data. The additional attempts here to standardize the measures may limit their reliability. This unfortunately limits the conclusions that can be drawn.

Our response: We thank the reviewer for noting the strengths of our study. We agree that there are challenges in harmonising data in order to undertake meta-analyses. We would like to note though that we have published metaanalysis of these cortisol data previously (Gardner et al 2013) and that we have been explicit about the decisions we made when cleaning the cortisol data. Our treatment of the cortisol data is typical of that for large, community-based studies (removing outliers and those on relevant medication, correcting for deviations from protocol in the timing of the sample) and we do not consider that this was influenced by standardising across the four cohorts. We justify our choice for each cortisol profile indicator included in the responses below. The figures show results from each individual cohort for completeness.

Gardner MP, Lightman S, Sayer AA, Cooper C, Cooper R, Deeg D, Ebrahim S, Gallacher J, Kivimaki M, Kumari M, Kuh D, Martin RM, Peeters G, Ben-Shlomo Y; Halcyon Study Team. Dysregulation of the hypothalamic pituitary adrenal (HPA) axis and physical performance at older ages: an individual participant meta-analysis. *Psychoneuroendocrinology*. 2013;38:40-9.

1) In both the CAPS and NSHD studies, salivary cortisol was measured at the time of awakening and 30 minutes later. This protocol was designed to measure the 'cortisol awakening response (CAR)' (the increase from the awakening sample to 30 minutes later) (for a review see Stalder et al., 2016). The CAR is a key variable in this field, representing a dynamic measure that has been related to various physiological and psychological conditions. That the 2 studies had both morning measures (awakening and awakening + 30 minutes) is a strength, yet in the current study, the authors simply average the 2 samples, which causes the loss of valuable information with regards to the CAR.

Our response: Thank you for this suggestion. We originally did look at the CAR but elected not to include it in this paper because it is particularly sensitive to sample timing which varied between studies. In addition, previous evidence indicates that both high and low CAR could be suboptimal. As an aside, we found no evidence of an association between CAR and wellbeing. (Please see figure below).

Reviewer: Indeed I agree that both a high and low CAR is suboptimal. Please see my response to point #2 below.

2) The authors may have decided to average the two morning samples (as mentioned above) in order to combine with the data from NCDS study, where cortisol was measured within 45 minutes after awakening. However, as we know from the CAR literature, if a sample is collected at awakening, or the peak around 30 min (after a steep rise) or at 45 minutes, after a decline again, very different measures are collected. To then combine this (from the NCDS study) with the average of awakening and awakening +30 from the CAPS and NSHD studies is a crude method of combining measures for a hormone that is very sensitive to sampling time. I would advice running analyses with only the data from the CAPS and NSHD studies using

the CAR measure (i.e. the dynamic increase from awakening to awakening + 30 minutes), even if this implies having a smaller sample size.

Our response: Please see response to 1). We note that Table 1 shows mean morning cortisol (based on averages of awakening and awakening + 30 mins) that are similar in the 3 birth cohorts (19.7, 22.9 and 21.3 respectively). The peak is typically measured at 30-45 minutes post-waking (Adam & Kumari 2009).

Adam EK, Kumari M. Assessing salivary cortisol in large-scale, epidemiological research. *Psychoneuroendocrinology* 2009;34:1423-36.

Reviewer: In the field of psychoneuroendocrinology, the reason for having 2 measures (awakening, awakening+30 minutes) or with an additional third measure (awakening+45 minutes) is to measure the dynamic cortisol changes during this period.

The authors state that the sample averages are the same across studies. However, the averages omit important information. Just as a hypothetical example, participant (A) may have an awakening cortisol at 10 nmol/L, which increases to 30 nmol/L at the awakening+30 measure. His average is 20 nmol/L. Participant (B) may have an awakening cortisol at 20 nmol/L, which remains flat at 20 nmol/L at the awakening+30 measure. They both have an average of 20 nmol/L, but participant (A) has a healthy profile, while participant (B) is suffering from burnout, which is why he has a flat profile. This would clearly not be captured by using an average.

The authors state that the objective of the study is "to test the hypothesis that cortisol patterns indicative of dysregulated HPA-axis functioning would be prospectively associated with poorer wellbeing at follow-up".

However, an average of the 2 morning cortisol measures (e.g. awakening and awakening +30) will certainly not capture dysregulations. Moreover, using the average contradicts what the authors report in the introduction "Large-scale epidemiological studies have measured salivary cortisol sampled several times during the course of a day to capture its rising levels during the awakening response". Again, by using an average, this 'rise' cannot be captured, and important information is lost.

A measure of change in cortisol is used in this study for the evening proportion, but it is unclear why it is not also used for the morning.

Also, whether a measure is taken at awakening+30 or awakening+45 does make a difference. Usually +30 is the peak, while +45 is the decline following the peak. In order to make it easier for the authors to visualize the point I am making, I'm attaching some examples from previous articles in the field.

Vreeburg, S. A., Hoogendijk, W. J., van Pelt, J., DeRijk, R. H., Verhagen, J. C., van Dyck, R., ... & Penninx, B. W. (2009). Major depressive disorder and hypothalamic-pituitary-adrenal axis activity: results from a large cohort study. *Archives of general psychiatry*, 66(6), 617-626.

Ulrike, S., Reinhold, L., & Dirk, H. (2013). Major depression in young girls is related to altered cortisol awakening response. *European child & adolescent psychiatry*, 22(6), 379-384.

3) Similarly, the authors state that they also combined the aforementioned cortisol samples with the serum cortisol measures from the HCS study, and cited one study showing the high correlations between both salivary and serum cortisol levels.

However, the morning serum cortisol (representing morning cortisol at one time point) which was measured hours after awakening, when participants went to the study site, is very different than the CAR measured in the CAPS and NSHD studies (representing a response to awakening). Combining them is likely to dilute potential existing associations.

Our response: We agree, it is possible that including HCS based on serum cortisol (and NCDS based on awakening + 45 mins) would dilute associations. However, as can be seen

in Figure 1A, there is no suggestion of an association between morning cortisol and wellbeing in NCDS, NSHD or HCS. There is a suggestion of an inverse association in CAPS. This is the one study that sampled cortisol over two days and so we make the recommendation that replication is needed with studies which have sample over multiple days.

Reviewer: This can be mentioned in the discussion.

4) For the CAPS and NSHD studies the diurnal slope was calculated by using the morning cortisol levels and the evening one. However, for the NCDS study, cortisol was measured in the first 45 min after awakening and 3 hours later. The authors justify using the measure 3 hours later by stating 'however past publications support the notion that the diurnal decline over this shorter period is a good surrogate for the decline from morning till night'. In fact they cite one study. The weakness of this method is that it assumes a linear decline to 3 hours and to the night cortisol levels, which is often not the case. I would suggest running the analyses with only the CAPS and NSHD studies, even if the sample size may be smaller.

Our response: As Figure 1C shows, the estimate of the association between diurnal slope and wellbeing was not materially different in NCDS and the other two studies included in that analysis. To address this comment, we have now repeated the meta-analysis including only NSHD and CAPS and we report this result in the text (page 12): "Results excluding NCDS (based on decline in cortisol between early and late morning) were similar (0.02 (95% CI - 0.41, 0.46))."

Reviewer: Thanks for now adding this.

5) It is unclear why only in the NCDS study, participants were excluded from analyses if they reported taking endocrine system medications, and not from the other studies.

Our response: We have taken the more conservative approach to exclude NCDS participants on endocrine system medication because we did not have access to more detailed data on cortico-steroid medication in this analysis.

Reviewer: Did you mean 'in this analysis' or from this dataset?

This should be mentioned as a limitation.

6) It is also unclear how the cut-off for salivary cortisol values was decided (100 nmol/L)

Our response: Examination of the salivary cortisol distribution indicated that values >100 nmol/L were clear outliers

Reviewer: That sounds reasonable

7) The authors state 'Morning salivary cortisol values that were not between 5am and noon were removed and evening values if they were before 8pm, since these participants may be shiftworkers with substantially different cortisol profiles'. Is there information available on whether or not individuals were shiftworkers? What proportion were they in the different populations? And as the authors state, since they have very different cortisol profiles, it would be important to perform sensitivity analyses with and without their values.

Our response: On reflection, given the age range of the cohort members, shiftwork is unlikely to be the primary explanation for morning values outside of 5am to noon and evening values before 8pm. We have edited the sentence on page 9 so that it now reads: "since these participants with atypical sleeping hours may be shift-workers with may have substantially different cortisol profiles."

Numbers were small (e.g. n=51 observations in NSHD) and so statistical power is too low to examine this group separately.

Reviewer: This modified sentence is unclear (grammar error).

Do the results change if these 51 are excluded?

8) In the HCS study, it is unclear what the time range and mean (SD) was for the fasting morning serum cortisol

Our response: We now clarify that the fasting morning serum cortisol sample was taken at a research clinic between 8.30 and 9.30 AM.

Reviewer: Thanks for adding this.

What is the mean (SD)?

9) It is unclear how participants were selected to participate in the cortisol sampling portions of the different studies

Our response: All those who had not withdrawn or been lost to follow-up at the relevant sweep were invited to participate in the cortisol sampling. This information has been added (page 7).

Reviewer: Thank you for adding this

10) In the introduction, the authors state 'The mental health and wellbeing consequences of raised cortisol levels are of interest particularly among older people given that they have higher evening cortisol levels'. The references provided are from 1996 and 2006. And indeed, this was believed to be the case. However, more recently it has been argued and shown that higher cortisol levels may indicate underlying pathology, whereas among healthy older adults, they do not show elevations in evening cortisol as a function of aging per se (see Souza-Talarico et al., 2011; Sindi et al., 2013).

Our response: Thank you for raising this. We have now clarified that there is inter-individual variation in how cortisol changes with age and the potential role of underlying pathology. The text (page 4) now states: "The mental health and wellbeing consequences of raised cortisol levels are of interest particularly among older people given that some studies find they have higher evening cortisol levels[1] and greater total cortisol output throughout the day[2] compared with younger people though there is inter-individual variation in age-related change in cortisol [3] and this may reflect age-related change in disease rather than normal ageing (new reference added Souza-Talarico et al. 2011)."

Souza-Talarico, J. N. D., Marin, M. F., Sindi, S., & Lupien, S. J. (2011). Effects of stress hormones on the brain and cognition: evidence from normal to pathological aging. *Dementia & Neuropsychologia*, 5(1), 8-16.

Reviewer: Thank you for adding this important point.

11) Regarding socioeconomic position, it was derived from own occupational class and grouped as manual or non-manual occupation. The limitation with this method is for example if women had manual occupations, yet their husbands had non-manual occupations, allowing for a high household income, then this would be an inaccurate measure. Can this variable be improved?

Our response: We checked whether own occupational class or head of household class was more closely related to cortisol and found a stronger association for the latter.

Reviewer: That is interesting. Did you incorporate it in the analyses?

12) Why was education not adjusted for?

Our response: We found no evidence of an association between education and cortisol when occupational class was adjusted for and so education did not meet the formal definition of a confounder

Reviewer: Thanks for your response.

13) Under 'initial treatment of the data and standardization', the authors state "Observed values were adjusted for the time of sampling by fitting a linear or polynomial function to the association between cortisol and time of measurement and adding the resulting residuals from the best fit model to the overall mean cortisol value. This gives the estimated cortisol level at the time specified in the protocol for each participant". Does this method take into account the consequences of deviation from sampling times, which are specific to the scheduled sampling time? What I mean is for example, a 15 min delay from awakening will lead to an-over-estimation of cortisol levels (because cortisol increases after awakening). In contrast, a 15 min delay for the awakening +30 minutes sample would lead to an under-estimation of cortisol levels (because cortisol decreases following the peak at awakening +30 minutes)

Our response: Yes, this approach accounts for the deviation from specific scheduled sampling time. The polynomial function allows for an average increase in cortisol with time before the specified time and an average decrease with time after the specified time, for example.

Reviewer: Thanks for your response

14) The descriptions of the different studies need elaboration. For example the follow-up time (range, mean and SD), participation rates, the mean age and SD at the different measurement time points, the numbers of participants in the main cohort study and those with measures available for the current study. Also, references are needed for previous publications where these cohort studies were described in more detail.

Our response: We have added the ranges, means and SD as suggested to Table 1. These cohort studies are described in more detail in the cohort profile papers (references 19-22).

Reviewer: Thank you for adding this information.

15) Under 'statistical methods', it states "cohort-specific estimates were pooled in random-effects meta-analyses".

However, the reference (#32) is an article titled

'Meta-analysis in clinical trials.' How is this reference on clinical trials relevant for the current study with cohort studies?

Our response: The random-effects meta-analysis approach is a well-accepted way of combining estimates of effect size across several studies and it is used for both randomised and non-randomised designs.(Here the "random-effects" term refers to the way that the estimates are combined across studies rather than to the original study design.) The authors of the paper we cited note this on page 186 of their paper. This is the standard reference for this approach. DerSimonian R, Laird N. Meta-analysis in clinical trials. *Control Clin Trials* 1986;7:177-88.

Reviewer: Thank you for the clarification.

16) Under results, the authors comment on the similarity between morning cortisol levels, but nothing is mentioned about the serum cortisol, night cortisol, or the slope.

Our response: We now note that night time cortisol was also similar but that there was some difference in slope for NCDS versus the other studies (page 11). As noted above, we now include analysis of diurnal slope with and without NCDS (page 12).

Reviewer: Thank you for adding this information.

17) When the authors state 'there was no evidence of heterogeneity across studies', it is unclear, heterogeneity in what?

Our response: We now clarify that there was no evidence of heterogeneity in the association between early morning cortisol and wellbeing across the studies (page 11).

Reviewer: Thank you. This is now clearer.

18) In Table 1, information is missing on the mean and SD under 'age range at cortisol measurement', similarly to how it was done for the WEMWBS row

Our response: We have now added mean and SD for age at cortisol measurement to Table 1

Reviewer: Thank you for adding this information.

19) In Table 1, information is need on follow-up time (range, mean, SD)

Our response: Mean and SD follow-up time has been added to Table 1.

Reviewer: Thank you for adding this information.

20) There is no information on participant consent, ethics approval etc

Our response: We now note that ethical approval was obtained for each of these studies. Rather than list the relevant ethics committee for each one, we refer again to the cohort profiles which describe these studies and the approval in more detail.

Reviewer: Thank you for adding this information. It is also standard for many journals to request for a statement that all participants provided written informed consent.

21) The following statement is currently under 'Initial treatment of the data and standardization', "In CaPS, NSHD and NCDS actual times when the salivary samples were taken were recorded by participants", instead it should be under 'Cortisol'.

Our response: Thank you for the suggestion but as this sentence is relevant for explaining how we derived cortisol level at the per protocol sampling time (rather than at the observed sampling time), we think it fits more appropriately in the section on initial treatment of the data.

Reviewer: Ok that sounds fine.

22) Minor comment: If the authors use the acronym BMI for body mass index, they should use it consistently in the manuscript.

Our response: We have replaced "body mass index" with BMI except for the first occurrence and where it appears in the Tables.

Reviewer: Thank you for making these edits.

Reviewer: Additional minor edits:

p.10 It states, 'we adjusted for sex and age at cortisol...', do you mean , age at cortisol

'measurements'?

p.13 The last part of this sentence needs revision, it is currently unclear: 'Only one of the included studies captured cortisol profiles on more than one day and this study only on two days'.

Warwick Edinburgh Mental Wellbeing Scale. The abbreviation was not defined before its first use in the abstract. In the manuscript, the abbreviation should be defined at its first appearance (p.5), and then used consistently across the manuscript (e.g. p.7 &14).

VERSION 3 – REVIEW

REVIEWER	Shireen Sindi Karolinska Institute, Sweden
REVIEW RETURNED	10-Jul-2017

GENERAL COMMENTS	I would like to thank the authors for thoroughly addressing the comments. I have no other comments to add. This will be an interesting contribution to the field.
---